# Tissue sodium excess is not hypertonic and reflects extracellular volume expansion

Giacomo Rossitto [1,2✉], Sheon Mary [1,2], Jun Yu Chen [1,2], Philipp Boder [1], Khai Syuen Chew [1], Karla B. Neves [1], Rheure L. Alves[1], Augusto C. Montezano[1], Paul Welsh[1], Mark C. Petrie[1], Delyth Graham[1], Rhian M. Touyz [1] & Christian Delles [1]

Our understanding of Na[+] homeostasis has recently been reshaped by the notion of skin as a depot for Na[+] accumulation in multiple cardiovascular diseases and risk factors. The proposed water-independent nature of tissue Na[+] could induce local pathogenic changes, but lacks firm demonstration. Here, we show that tissue Na[+] excess upon high Na[+] intake is a systemic, rather than skin-specific, phenomenon reflecting architectural changes, i.e. a shift in the extracellular-to-intracellular compartments, due to a reduction of the intracellular or accumulation of water-paralleled Na[+] in the extracellular space. We also demonstrate that this accumulation is unlikely to justify the observed development of experimental hypertension if it were water-independent. Finally, we show that this isotonic skin Na[+] excess, reflecting subclinical oedema, occurs in hypertensive patients and in association with aging. The implications of our findings, questioning previous assumptions but also reinforcing the importance of tissue Na[+] excess, are both mechanistic and clinical.

[1] Institute of Cardiovascular and Medical Sciences, University of Glasgow, Glasgow, UK. [2]These authors contributed equally: Giacomo Rossitto, Sheon Mary, Jun Yu Chen. ✉email: Giacomo.Rossitto@glasgow.ac.uk

I n the last decade, the concept of tissue Na$^+$ shifted the traditionally nephrocentric view of Na$^+$ homoeostasis to the interstitium[1,2], where the negatively charged glycosaminoglycans and a TonEBP/NFAT5-VEGF-c-mediated expansion of the lymphatic network serve as a depot and draining system for its accumulation, respectively[3–5]. The proposed water-independent nature of this phenomenon[6–8] marked its novelty[9,10] and its potential implications in the pathogenesis of hypertension and cardiovascular disease at large, as independent studies reported a boosted activation of pathogenic immune-inflammatory cells upon culture in supraphysiological concentrations of Na$^+$ [11–15]. The translational value of this evidence relies on the presence of hypertonic (HT) microenvironments in tissues. In humans, increased tissue Na$^+$ concentration has been identified in association with aging and hypertension[16], diabetes[17], chronic kidney disease[18] and acute heart failure[19] in skin (where the accumulation was initially described as specific in rodent models[3,7]) and/or skeletal muscle, as well as in sclerodermic[20] or infected skin[12], by means of $^{23}$Na-MRI. Excess Na$^+$ accumulation upon salt loading, with some sex specificities, was also found in healthy subjects by direct skin chemical analysis[21]. Importantly, both methods measure Na$^+$ in the whole tissue. We have previously suggested that tissue architecture per se can markedly impact on Na$^+$ content and concentration (Fig. 1)[22]. Such histochemical[23,24] considerations are key to conclude on the HT nature of this Na$^+$ excess, which currently lacks demonstration in humans. In fact, even in rodent models, attempts to isolate hyperosmolar interstitial fluid or lymph proved unsuccessful[25]. Moreover, the dual osmotically active and inactive nature of interstitial Na$^+$, driving TonEBP-mediated signalling while simultaneously eluding parallel and commensurate water accrual, appears equivocal. On these premises, we sought to test the existence, distribution and putative correlates of HT tissue Na$^+$ accumulation by probative and disprobative approaches (i.e., by verifying and by assuming its occurrence, respectively) in preclinical models of and real-life patients with arterial hypertension.

Our study offers a reappraisal of the tissue Na$^+$ theory, disproves its water-independence in both experimental salt-sensitive hypertension and hypertensive subjects and suggests systemic isotonic (IT) Na$^+$ excess as an important player in the pathogenesis of cardiovascular disease, particularly in association with the process of aging.

## Results

**Tissue Na$^+$ accumulation is systemic and isotonic.** Male and female stroke-prone spontaneously hypertensive (SHRSP) and control Wistar–Kyoto (WKY) rats (12 weeks old) were treated with 1% NaCl in drinking water (high salt, HS) or normal tap water (normal salt, NS) for 3 weeks. In males, baseline blood pressure (BP) was higher, irrespective of strain differences, and salt sensitivity of BP was confined to SHRSPs (Supplementary Fig. 1). Skin Na$^+$ content (mmol g$^{-1}$ of dry weight (DW)) was increased in SHRSP-HS (+20.7%), as expected; however, we observed similar increases in tissue Na$^+$ also in liver (+10.3%), lungs (+14.8%) and skeletal muscle (+23.6%), but not in myocardium (Fig. 2a and Supplementary Data 1). Apart from skeletal muscle, they were consistently paralleled by increases in tissue water (skin: +14.1%; liver: +4.8%; lungs: +9.0%) with similar trends observed for Na$^+$ concentrations (i.e., normalised for water content, [Na$^+$], mmol L$^{-1}$; Supplementary Data 1). These changes upon HS were not observed in WKY rats. A sub-analysis for males and females showed almost identical patterns (Supplementary Data 1). Different experimental results for tissue Na$^+$ across organs fit well with a prediction model based on the intracellular/extracellular volume proportion (ECV/ICV) typical

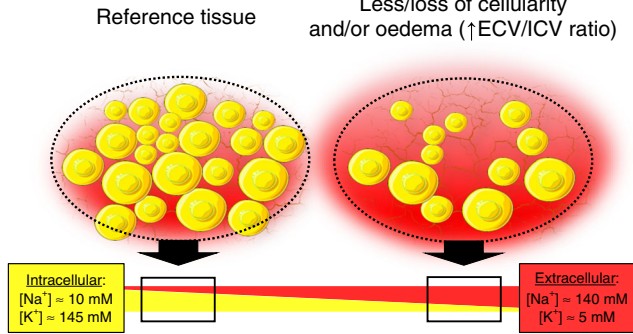

## (HISTO) Chemical analysis — [Na$^+$] and [K$^+$]

Reference tissue | Less/loss of cellularity and/or oedema (↑ECV/ICV ratio)

Intracellular: [Na$^+$] ≈ 10 mM [K$^+$] ≈ 145 mM

Extracellular: [Na$^+$] ≈ 140 mM [K$^+$] ≈ 5 mM

## $^{23}$Na Magnetic resonance imaging

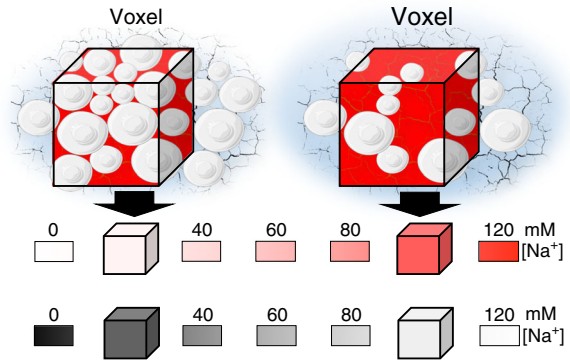

Voxel                          Voxel

0   40   60   80   120 mM   [Na$^+$]

0   40   60   80   120 mM   [Na$^+$]

**Fig. 1 Histochemical impact of tissue architecture.** Intracellular compartments are rich in K$^+$ (yellow), while extracellular compartments in Na$^+$ (red); in both, the sum of K$^+$ and Na$^+$ concentrations ([K$^+$] and [Na$^+$], respectively) is similar. When intracellular and extracellular compartments are mixed, i.e. a whole tissue chemical analysis is performed, total [Na$^+$] is higher in any tissue with physiologically less intracellular volume or undergoing loss of cellularity or oedema accumulation compared to a reference tissue. The concept applies to $^{23}$Na-MRI (bottom), where the average signal from a voxel is presented as a shade of colour, typically grey in the commonly used radiological black–white lookup table. ECV extracellular volume, ICV intracellular volume.

of each tissue (Supplementary Fig. 2a)[22,26]. Lower water and Na$^+$ between control (NS) strains in skin and liver reflect differences in fat content that limit the volume of distribution (and total content, accordingly) of both (Supplementary Fig. 3). Importantly, K$^+$ concentration ([K$^+$]) showed opposite trends to [Na$^+$] (Supplementary Data 1). In physiological conditions, the sum of [K$^+$] and [Na$^+$] in both ECV and ICV is similar, although with opposite predominance of Na$^+$ and K$^+$, respectively; a mixture of ECV and ICV in any proportion would still result in a solution of ~140–160 Na$^+$ + K$^+$ mmol L$^{-1}$. In all tissues, [Na$^+$ + K$^+$] consistently fell within this range, with no differences between strains or NS/HS allocation (Fig. 2a). Overall, rather than any HT phenomenon, this points to a shift in ECV/ICV ratio and, along with the concomitant increase observed for tissue water, to systemic oedema. The difference in the proportional changes of Na$^+$ and water does not reflect different amounts, as previously suggested, but different sensitivities to oedema detection (Supplementary Fig. 2b)[22]. In fact, water paralleled Na$^+$ content with identical regression slopes across all groups (Supplementary Fig. 4), except for skeletal muscle where the water-independent ECV/ICV shift is likely due to HS-induced sarcopenia[27]. The lack of oedema in skeletal muscle and myocardium also reflects the lower physical

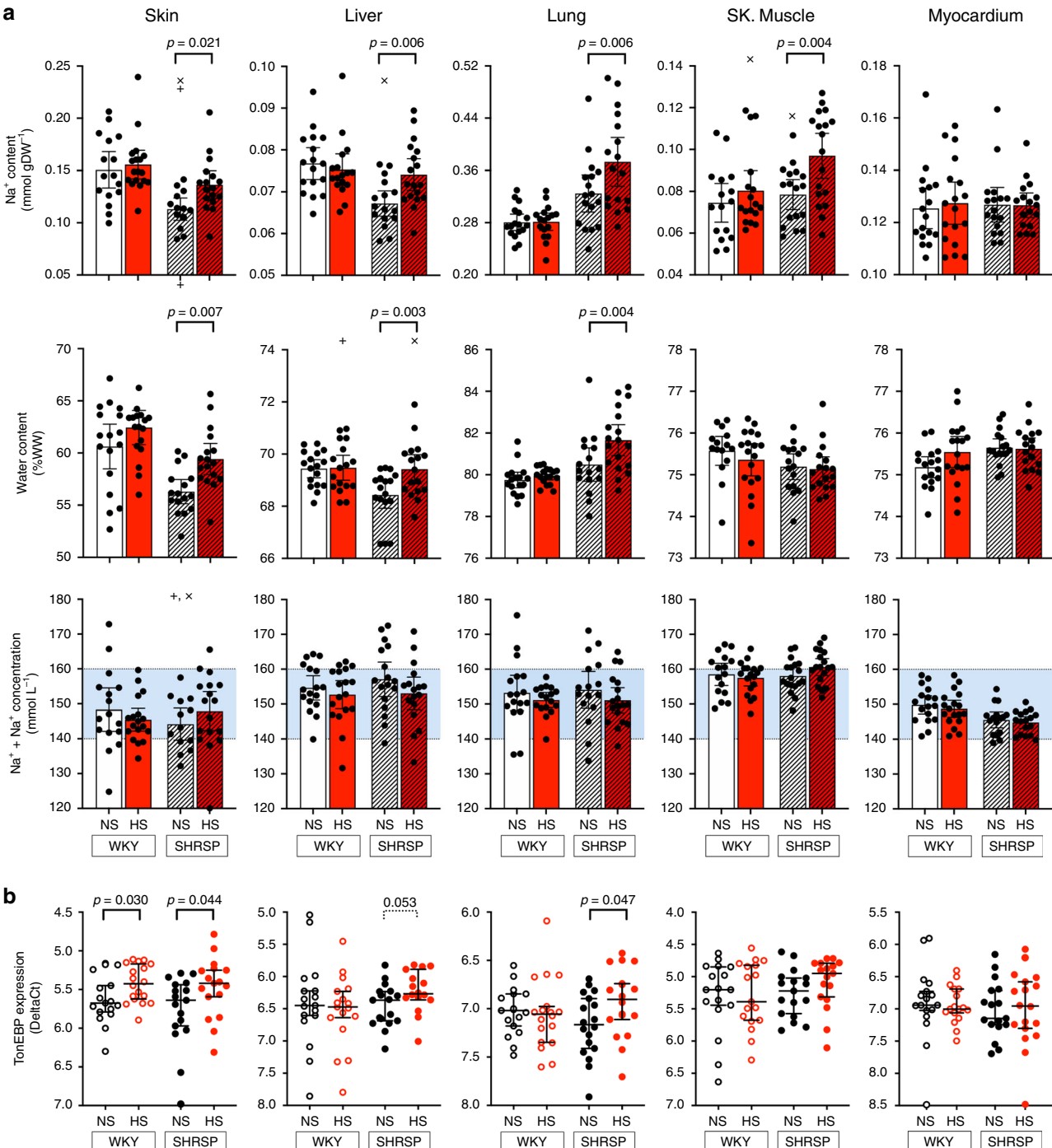

**Fig. 2 Nature and distribution of tissue Na⁺ excess and TonEBP expression upon salt loading.** Panel **a** bars: empty = *WKY*, dashed = *SHRSP*, red = high salt (HS) treatment; all bars show mean ± 95% CI, with individual points (overlay; + and x: female and male automatically detected outliers, respectively; ROUT, Q = 1%); significant differences at predefined comparisons are shown by brackets, on top (two-sided Fisher LSD test; please see also Supplementary Table 1 for sex-specific data and detailed significance). Tissue Na⁺ content increased in all tissues in *SHRSP*-HS, but in myocardium; with the exception of skeletal muscle, it was consistently paralleled by water accrual. Total Na⁺ + K⁺ concentration in tissues fell in physiological ranges (light blue) and was unaffected by salt loading in either strain, ruling out hypertonic accumulations. Panel **b** data presented as Delta Ct, median ± 95% CI for all rats; significant differences at two-sided Mann–Whitney test are shown in brackets. Increased *TonEBP* gene expression followed Na⁺ and water accumulation despite no hypertonicity. For both panels, n = 16–18 animals/group, as per Supplementary Data 1; source data are provided as a Source Data file.

compliance of these tissues[28]; of note, in similar experiments conducted in NS-fed but aged rats (52 weeks old), myocardial oedema was detectable and accompanied by extracellular matrix remodelling with an increase in glycosaminoglycans (Supplementary Fig. 5).

The currently proposed mechanistic framework indicates that water-independent tissue Na⁺ excess is responsible for a VEGF-c mediated signalling via TonEBP/NFAT5[3,4]. Although traditionally associated with the response to osmotic stress[29], we observed an increase in *TonEBP/NFAT5* expression in skin, lungs and

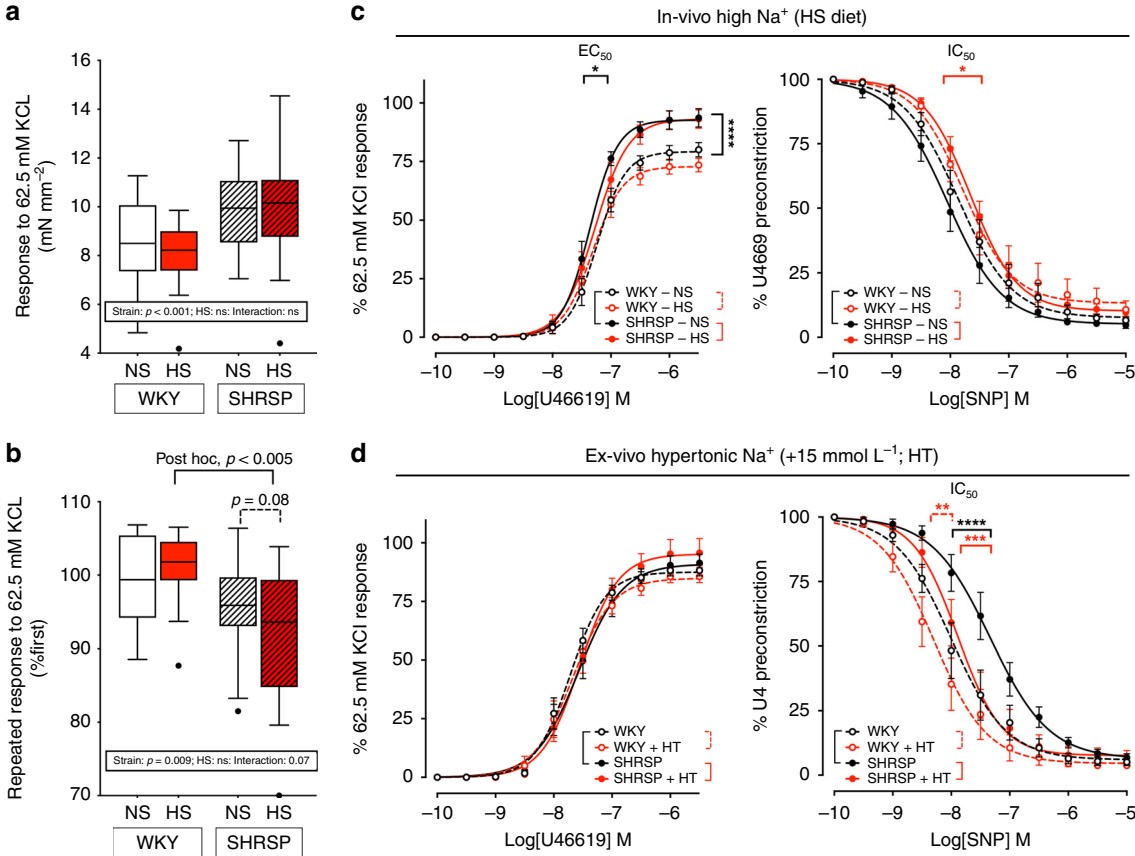

**Fig. 3 Impact of in vivo and ex vivo Na$^+$ excess on vascular function of resistance arteries.** Panels **a**, **b** contractile response to KCl, by strain and experimental treatment; box-and-whisker plots (Tukey: centre line, median; box limits, upper and lower quartiles; whiskers, 1.5x interquartile range; points, outliers); WKY-NS: $n = 15$ and HS $n = 18$, SHRSP-NS: $n = 14$ and HS $n = 17$; two-way ANOVA results at bottom and predefined or post hoc (inter-strain) comparisons on top. Panel **a** the response to the first KCl stimulation was higher in SHRSP than WKY but was unaffected by in vivo treatment; panel **b** the response to a second stimulation, expressed as percentage of first peak response, was reduced in SHRSP, particularly upon HS. Panels **c**, **d** contraction and relaxation dose–response curves to U46619 (thromboxane A$_2$ receptor agonist; $n = 11$–17/group, please see Supplementary Table 1a, b) and SNP (endothelium-independent nitric oxide donor; $n = 9$–16/group, please see Supplementary Table 1a, b), respectively; mean ± SEM. Differences in maximal responses and half maximal effective/inhibitory concentrations (EC$_{50}$ and IC$_{50}$, respectively) across groups are shown by brackets; *$p < 0.05$, **$p < 0.01$, ***$p < 0.001$, ****$p < 0.0001$, at Extra sum-of-squares F test upon non-linear regression by least-square method. **c** Effect of 1% NaCl in drinking water (HS): HS did not worsen the hypercontractile phenotype observed in SHRSP rats (max response to U46619: SHRSP-NS vs. WKY-NS, $p < 0.0001$; SHRSP-NS vs. SHRSP-NS, $p = 0.892$; LogEC$_{50}$ to U46619: SHRSP-NS vs. WKY-NS, $p = 0.017$; SHRSP-NS vs. SHRSP-NS, $p = 0.142$), but reduced their sensitivity to NO-mediated relaxation (max response to SNP: SHRSP-NS vs. SHRSP-NS, $p = 0.041$; LogIC$_{50}$ to SNP: SHRSP-NS vs. SHRSP-NS, $p < 0.0001$); no significant effect was observed in WKY rats. **d** Effect of ex vivo 5 h incubation in a hypertonic (+15 mmol/l NaCl, HT) vs. physiological solution on vessels from NS rats: ex vivo hypertonic Na$^+$ excess did not affect contractile responses (max response to U46619: SHRSP-HT vs. SHRSP, $p = 0.328$; WKY-HT vs. WKY, $p = 0.382$; LogEC$_{50}$ to U46619: SHRSP-HT vs. SHRSP, $p = 0.856$; WKY-HT vs. WKY, $p = 0.516$), but induced oversensitivity to NO in both strains, opposite to in vivo HS (LogIC$_{50}$ to U46619: SHRSP vs. WKY, $p < 0.0001$, SHRSP-HT vs. SHRSP, $p < 0.0001$; WKY-HT vs. WKY, $p = 0.007$); none of these ex vivo effects would justify an increase in peripheral vascular resistance. Please see Supplementary Table 1a, b for all comparisons; source data are provided as a Source Data file.

(borderline significant) liver in SHRSP-HS, as well as in myocardium of aged SHRSP, despite the lack of hypertonicity (Fig. 2b and Supplementary Fig. 5). As recently demonstrated for VSMCs[30], we speculate that biomechanical (rather than osmotic) stress due to oedema accumulation is responsible for the activation of the lymphangiogenic signal cascade.

**Extracellular hypertonic Na$^+$ does not induce hypertensive vascular dysfunction.** Blockade of lymphangiogenesis induces development of salt-sensitive hypertension[3,4]. If the resulting excess tissue Na$^+$ were independent of water (i.e., volume), the expectation is that it would impact on the other determinant of hypertension, i.e., peripheral resistance. To test this hypothesis, we investigated the vascular function of peripheral resistance arterioles from our NS/HS rats. HS treatment did not result in

hypercontractile responses to U46619 (a thromboxane-receptor agonist) or KCl but reduced the latter upon repeated stimuli and the sensitivity to the vasorelaxant effect of nitric oxide (NO) in SHRSP (Fig. 3 and Supplementary Table 1). Intriguingly, the concept that vascular swelling could determine increased peripheral resistance and vascular stiffness (to both contraction and relaxation) in the pathogenesis of hypertension dates back to the 1950s[31]; although it is supported by subsequent evidence[32–34], which extends to related diseases[35] and includes preliminary data from our rats (Supplementary Fig. 6), it still lacks firm demonstration. To test whether an HT environment could directly induce similar changes, arterioles from NS rats were also preincubated in an IT or HT (+15 mmol L$^{-1}$ NaCl) solution. Akin to HS in vivo, HT ex vivo incubation did not affect contractile responses; however, it induced an opposite shift toward an earlier

NO-induced loss of pre-constriction tone in both *WKY* and *SHRSP* (Fig. 3 and Supplementary Table 2).

This is not reminiscent of classic hypertensive vascular phenotypes and appears more consistent with the neuromuscular signs of hypernatremia (e.g., lethargy and muscle weakness). Overall, these data suggest that impaired drainage of tissue $Na^+$ is unlikely to induce hypertension by directly and adversely modulating peripheral vascular function in a water-independent manner.

**Skin $Na^+$ accumulation in hypertensive patients.** To translate our preclinical findings to humans, we performed a chemical analysis of arm skin punch biopsies of 76 hypertensive subjects (Supplementary Table 3). The two anatomically distinct layers of epidermis/superficial dermis (ESD) and deep dermis (DD) were analysed separately: as our histochemical approach would predict, $[Na^+]$ and $[K^+]$ were lower and higher in the ESD compared to DD, respectively (Fig. 4a). In virtually no patients, and thereby in none of the prespecified different clinical subgroups, did $[Na^++K^+]$ exceed the physiological value of 155 mmol $L^{-1}$ in either layer (Fig. 4b), thus ruling out any HT accumulation. In

both ESD and DD, water content was positively and negatively correlated with $[Na^+]$ and $[K^+]$, respectively (Fig. 4c); in DD, water, $Na^+$ and $K^+$ contents, but not concentrations, were all positively and highly correlated (Fig. 4c), primarily reflecting their volume of distribution which excluded dermal fat. Accordingly, male DD contained more water, $Na^+$ and $K^+$ than female dermis, which is known to be richer in subcutaneous fat (Supplementary Fig. 7); no such differences were seen in the ESD (Supplementary Table 4). Identical histochemical sex differences were observed in gluteal biopsies from young healthy volunteers (Supplementary Table 5); notably, fluid-retentive[36] progestinic states were characterised by higher water and $Na^+$ content, but unchanged $[Na^+ + K^+]$ (Supplementary Fig. 7). In the hypertensive cohort, age was independently associated with an increase in ESD water and $[Na^+]$, and a decrease in $[K^+]$ in both layers (Fig. 4d and Supplementary Fig. 8); this likely indicates excess fluid accumulation in the context of reduced tissue cellularity, traditionally accepted as a hallmark of skin aging (Fig. 4d). Salt intake predicted epidermal water content, but not $Na^+$ or $K^+$, independently of age, sex and BMI (Supplementary Fig. 8); at variance

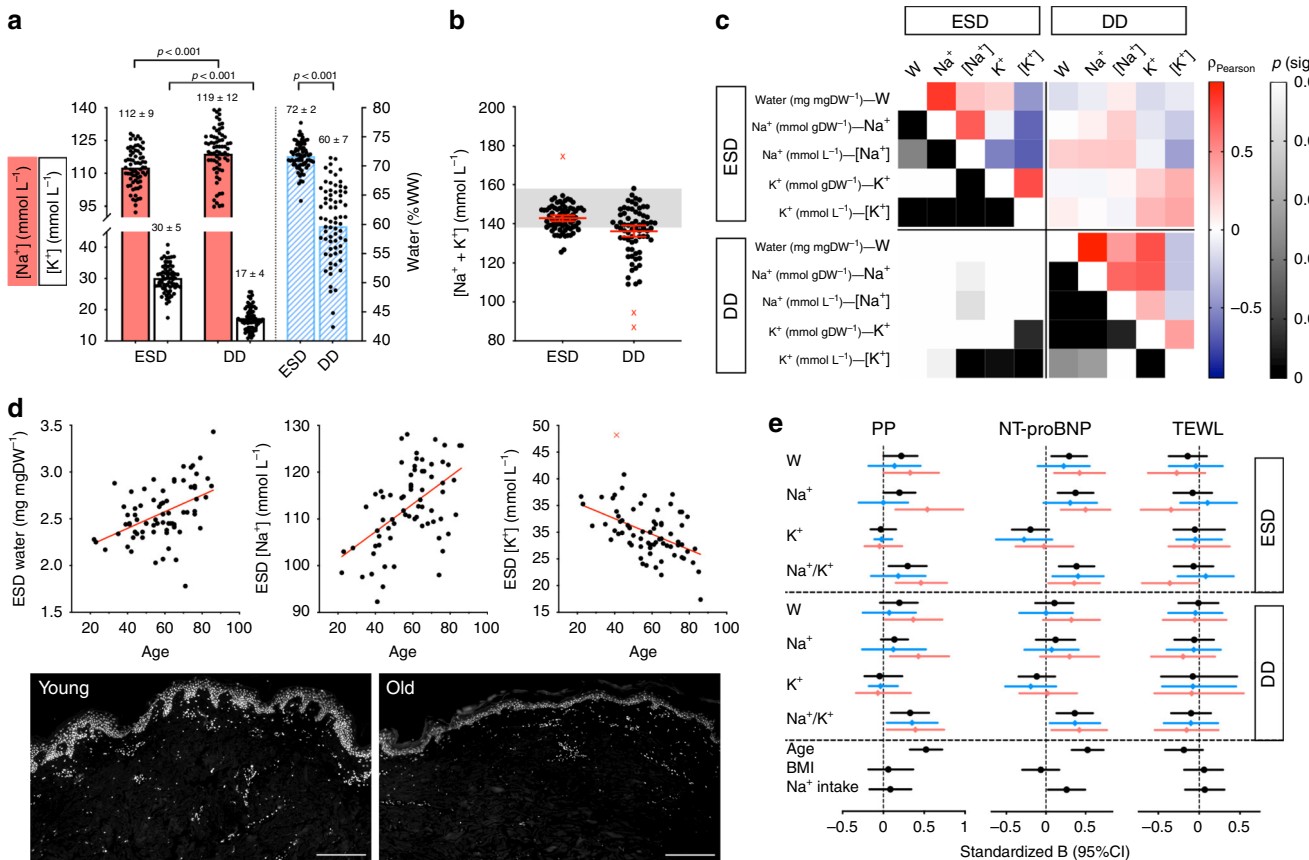

**Fig. 4 Histochemical analysis of skin from hypertensive patients and clinical correlates.** Panel **a** bars show mean ± 95% CI, with individual points (overlay); $Na^+$ and $K^+$ concentrations and water content in the epidermis/superficial dermis (ESD) and deep dermis (DD) layers, reflecting different architecture and cellularity. Panel **b** skin $Na^+ + K^+$ concentration in ESD and DD; red X = automatically detected outliers (ROUT, $Q = 1\%$); in virtually no patient, including $n = 3$ cases with primary aldosteronism, did $[Na^+ + K^+]$ exceed physiological values (grey). Panel **c** correlations of histochemical parameters in ESD and DD; Pearson $\rho$ and statistical significance are colour coded as per legend. Panel **d** association of ESD water, $[Na^+]$ and $[K^+]$ with age, reflecting the development of oedema and the shift in ECV/ICV ratio; red X = automatically detected outlier; bottom: representative pictures of age-related reduction in skin cellularity; immunofluorescence, DAPI staining; scale bars (bottom right) = 200 μm. Panel **e** relationships between skin histochemical parameters, age, BMI, Na intake and pulse pressure (PP), plasma NT-proBNP and transepidermal water loss (TEWL), by skin layer. Data are presented as standardised B regression coefficients (95% CI); for skin histochemical parameters, coefficients for all patients are visualised in black, for males in blue, for females in pink. PP and NT-proBNP positive associations with ESD water accumulation were mostly driven by females; the association with $Na^+/K^+$ ratio was overall generalised to both layers and sexes. No significant associations were observed with TEWL, making relative local water deficit an unlikely explanation for any $Na^+$ hypertonic excess. All panels refer to $n = 76$ samples from independent subjects; missing data due to technical issues or unavailable samples were ≤8 for all variables/correlations; source data are provided as a Source Data file.

with aging, the lack of any significant shift in the Na/K ratio in relation to salt intake would suggest a fluid accumulation evenly distributed between ICV and ECV. Office diastolic BP was inversely associated with ESD water content ($\rho = -0.250$; $p = 0.03$) and a surrogate of vascular stiffness, pulse pressure (PP), was positively correlated with $Na^+/K^+$ ratios (Fig. 4e); in ESD, the ratio was higher in patients with uncontrolled BP, independent of other covariates. No significant interaction with medications was observed. NT-proBNP levels, positively associated with age ($\rho = 0.526$; $p < 0.001$), estimated $Na^+$ intake ($\rho = 0.260$; $p = 0.03$) and systolic BP ($\rho = 0.288$; $p = 0.01$) and negatively with diastolic BP ($\rho = -0.277$; $p = 0.02$), mirrored all measures of skin oedema accumulation (Fig. 4e). Finally, the lack of correlations between histochemical skin data and transepidermal water loss (TEWL)[37] (Fig. 4e) stood against the predefined hypothesis that local regulatory mechanisms could make skin an exception to the IT nature of systemic tissue $Na^+$ excess/accumulation via a relative deficit of water.

## Discussion

Our study has epistemological, mechanistic, diagnostic and therapeutic implications. First, while we cannot unconditionally exclude an HT $Na^+$ accumulation in any tissue milieu or any clinical/experimental condition, our findings prompt reconsideration of excess tissue $Na^+$ epidemic in light of a simpler histochemical deductive approach[24]. Numquam est ponenda pluralitas sine necessitate[38] (plurality is not to be posited without necessity). Over 80 years ago, Simms and Stolman made cadaveric observations consistent with our conclusions[39]. Second, this concept of tissue $Na^+$ accumulation is not herein disproved but reinforced and reinterpreted, confirming the importance of regulatory mechanisms like TonEBP/NFAT5 signalling and lymphatic network plasticity in systemic fluid homoeostasis. They should indeed be a focus for future research, together with the biophysical/mechanical impact of cellular rarefaction and clinically overt or subclinical oedema, rather than hyperosmotic niches, on organ function. Third, such foreseeable impact could be tracked, by non-invasive tissue $Na^+$ analysis, in many relevant clinical scenarios beyond sole cardiovascular disease. Finally, the effect of novel natriuretic agents in both prevention and treatment of major cardiovascular events[40,41] prompts better identification of those patients who will benefit most. Considering the demonstrated potential of these agents in reducing tissue $Na^+$ content[3] and the above functional considerations, it is tempting to speculate that this quest should take into account systemic and/or organ-specific tissue $Na^+$ analysis.

## Methods

**Animals**. All protocols were performed in accordance with the United Kingdom Animals Scientific Procedures Act 1986 (Project Licence 70/9021 held by Delyth Graham) and ARRIVE Guidelines and approved by the institutional ethics review committee (University Animal Welfare and Ethics Review Board, University of Glasgow, Glasgow, UK).

We used SHRSP and WKY rats from colonies inbred at the University of Glasgow since 1991; animals were housed under controlled environmental temperatures ($21 \pm 3\,°C$) and lighting (12-h light–dark cycles).

In the main set of experiments, male and female SHRSP and WKY were maintained on standard rat diet (rat and mouse No. 1 maintenance diet; Special Diet Services, Grangemouth, United Kingdom) and provided tap water ad libitum until 11 weeks of age. At 12 weeks, rats (littermates) were randomised to 1% NaCl (HS) or normal drinking water (NS) for 3 weeks ($n = 8$ males/group and $n = 10$ females/group, except $n = 9$ female WKY-NS). Systolic BP was measured at 11 weeks of age (baseline) and then monitored weekly by tail-cuff plethysmography, in an operator-blind fashion whenever possible. In another set of experiments, male SHRSP and WKY were maintained on standard diet and normal tap water ad libitum until 20 and 52 weeks of age (WKY 20 weeks $n = 10$; WKY 52 weeks = 9; SHRSP 20 weeks = 6; SHRSP 52 weeks = 9).

**Tissue harvesting and processing**. At the end of the 3 weeks of experimental treatment (main experiment) or at the appropriate age (additional experiment) rats were euthanized by exsanguination under general terminal isoflurane anaesthesia. Myocardium (left ventricle), lungs, liver, descending aorta and samples from shaved abdominal skin and skeletal muscle from the thigh, freed of fascia and tendon, were dissected and snap-frozen in liquid nitrogen. The dissection time was kept to a minimum to prevent evaporation of moisture. Any excess blood was removed by gently blotting the tissues on tissue paper; prior to blotting, heart was cut into transverse sections and aorta lumen gently washed by gravity with a phosphate buffer saline-filled syringe. An aliquot from each tissue was cut on the surgical table and stored into tubes pre-filled with RNA-later (Qiagen Group). All tissues were stored at $-80\,°C$ until use.

From frozen tissue samples, aliquots representative of full parenchyma were cut and their wet weights (WW) were measured on a four decimal (0.0001 g) electronic scale (samples WW range $= 15$–$60\,μg$). To prevent evaporation of tissue water, cutting was performed in a cold room for all the main set of experiments and tissues were transported in eppendorf tubes (Sarstedt) in dry ice. Samples were desiccated at $65\,°C$ in a ThermoMixer (Eppendorf), for $>40$ h, to a stable DW. Water content was estimated as $(WW - DW)/DW$ and expressed as mg water/mgDW, or as water percentage ($W\% = (WW - DW) \times 100/WW$). Dried samples were digested at $65\,°C$ in 20–40 μl of 70% $HNO_3$ (Fisher) for 3 h and, after 1:10 v/v dilution in deionised water (MilliQ), for 2 additional hours. Digestion blank controls (i.e., $HNO_3$ and MilliQ only) were prepared and analysed with the digested samples to confirm the lack of $Na^+/K^+$ contaminations. Digested samples and blanks were centrifuged for 1 min at $16,000 \times g$ and stored at room temperature; solubilised supernatants were used for $Na^+$ and $K^+$ quantification, while acid-insoluble residues (present in significant amount in skin and liver samples) were used for total fat quantification.

**$Na^+/K^+$ measurement**. $Na^+$ and $K^+$ in the digested samples were quantified by flame photometry (Sherwood scientific, model 410 C). $Na^+$ and $K^+$ calibration standards of 0–5 ppm (mg/l) were prepared from 1000 ppm solutions (Fisher Chemical) using $HNO_3$– and milliQ– carefully washed glassware. Before photometric measurements, digested samples were further diluted to fall within the range of 0–5 ppm. MilliQ was used as the diluent for both standards and samples after $Na^+/K^+$ contamination of $HNO_3$ was excluded on blanks. $Na^+$ and $K^+$ concentrations in the digested sample solutions were calculated against the five-points regression line obtained from the calibration standards. At the calibration used, reported CV% for reproducibility is <2% (i.e., <0.005 mmol/l for both Na and K in the measured sample); in our hands CV% was 1.5 intra-sample and 3% inter-sample (from the same original tissue). Concordance correlation coefficient for replicated (technical) measurements was 0.98 (Supplementary Fig. 9a); random duplicate samples from the same stored tissues showed similarly good reproducibility (Supplementary Fig. 9b, c). All samples from the same type of tissue were analysed in a batch on the same day to minimise technical variability; therefore, few samples irreversibly affected by experimental issues (e.g., accidentally dropped sample, contamination, volume shortage) had to be excluded and are reported as blank cells in the Source Data File. During measurements, blank (diluent) and calibration standards were checked after every block of approximately eight samples to control for drift. $Na^+$ and $K^+$ concentrations in the analysed solutions were used to back-calculate their total content in the digested samples and normalised by DW for $Na^+$ and $K^+$ tissue content (mmol/gDW), or by tissue water for $Na^+$ and $K^+$ tissue concentration (mmol/l). During flame photometry analysis, operators were blind to group allocation of samples.

**Fat content analysis**. After careful removal of the solubilised material used for $Na^+/K^+$ quantification, total fat was extracted from the acid-insoluble residue of skin and liver samples in 200 μl of a solvent mixture (chloroform/methanol, 2:1 v/v; protocol adapted from Lowry et al.[24] and Folch et al.[42]); extraction of the total fat prevented replication of this specific experiment. After overnight incubation at room temperature under agitation, solubilised extracts were transferred to pre-weighed glass tube inserts (Agilent Technologies) and solvents evaporated at $55\,°C$ to constant weight through repeated SpeedVac cycles. Total fat content in tissues was normalised by original WW (%WW).

**Histology**. Sulphated glycosaminoglycans (sGAG) myocardial content was estimated by Alcian Blue staining of available paraffin-embedded transverse mid-ventricular 5 μm sections from the 20- or 52-week-old WKY and SHRSP male rats ($n = 4$–5/group). After deparaffinization (Histoclear) and progressive rehydration to distilled water, slides were incubated in Alcian Blue solution (1% Alcian Blue 8GX, Sigma, in 3% acetic acid; pH = 2.5) for 20 min, washed in running tap water for 5 min, rinsed in distilled water and counterstained in Nuclear Fast Red solution (Vector), washed again in running tap water for 1 min and mounted following dehydration.

Random, non-consecutive pictures were taken at ×40 from mid-myocardium ($n \geq 25$/animal) and subepicardium ($n \geq 12$/animal; immediately below the epicardial layer) with an Olympus BX41 microscope and dedicated image capture software. sGAG were quantified as %AlcianBlue+ stained area after standardised

thresholding across pictures, by use of an in-house developed macro (Supplementary Note 1) for ImageJ.

**TonEBP gene expression analysis**. For all tissues, total RNA extraction was performed using Qiazol (Qiagen) and RNAesy mini-column kit (Qiagen) according to manufacturer's guide. For qRT-PCR, CDNA was prepared using the High capacity CDNA reverse transcription kit (Applied Biosystems) and analysed using Taqman fast advanced master mix with specific Taqman gene expression assay probes for *TonEBP* (Rn01762487_m1), *GAPDH* (Rn01462661 g1), *beta-actin* (Rn00667869 m1). The expression levels were normalised to either *GAPDH* (myocardium) or *beta-actin* (other tissues; housekeeping gene was selected upon evidence of Ct consistency across groups) and were compared and presented as delta Ct values (inversely proportional to gene expression level).

**Rat vascular function studies**. After organ harvesting for tissue chemical/molecular analysis, mesenteries were also dissected from NS/HS *WKY* and *SHRSP* rats and small resistance arteries were isolated as previously described[43]. Briefly, arterial segments were mounted on isometric wire myographs (Danish Myo Technology, Denmark) filled with 5 ml of physiological saline solution (PSS; 119.0 mM NaCl, 4.7 mM KCl, 1.2 mM $MgSO_4 \cdot 7H_2O$, 24.9 mM $NaHCO_3$, 1.2 mM $KH_2PO_4$, 2.5 mM $CaCl_2$ and 11.1 mM glucose) and continuously gassed with a mixture of 95% $O_2$ and 5% $CO_2$ while being maintained at a constant temperature of $37 \pm 0.5\,°C$. LabChart (ADinstruments) was used for data recording. Following 30 min of equilibration, baseline tension was normalised as per DMT recommendations and internal diameter was estimated (https://www.dmt.dk/uploads/6/5/6/8/65689239/dmt_normalization_guide.pdf); vessels with an internal diameter $\geq 500\,\mu m$ were excluded from further analysis. After normalisation, the contractility of arterial segments was assessed by the addition of KCl (62.5 mmol/L), repeated after 10 min of washout. The contractile response of vessels from the four experimental groups to the thromboxane agonist U46619 was tested with concentration-response curves ($10^{-10}$ to $3 \times 10^{-6}$ M); endothelium-independent vasorelaxation was assessed by a dose–response to the NO-donor sodium nitroprussiate (SNP; $10^{-10}$ to $10^{-5}$ M), following pre-constriction with the U46619 dose that produced 75% of the maximal contractile response. Other vessels from NS-treated animals were incubated for 5 h in PSS or in NaCl-supplemented-PSS (+15 mM NaCl, HT) to test the impact of environmental hypertonicity on vascular function with an ex vivo approach. Duration and tonicity of the incubations were based on previously reported tissue changes with HS diet[3,4] and effects of HT culture conditions on rat VSMCs hypertrophy[44]. After the incubation and one additional KCl stimulation, concentration-response curves for U46619 and SNP were conducted as described above. Vessels with no or trivial response to KCL or U46619 pre-constriction were considered non-viable and their curves were excluded from the analysis.

**Human hypertensive subjects**. The protocol for the cross-sectional $S_2ALT$ (Skin Sodium Accumulation and water baLance in hypertension) study was approved by the West of Scotland Research Ethics Committee 3 (ref. 18/WS/0238) and Greater Glasgow and Clyde NHS Research and Development (ref. GN18CA634). The study was conducted in compliance with the Declaration of Helsinki. Adult, non-pregnant patients were recruited from the High BP clinic, Queen Elizabeth University Hospital, Glasgow between March and July 2019. On the day of their scheduled clinic appointment (9.00 a.m. to 4.30 p.m.), non-pregnant patients willing to take part gave written, informed consented and had anthropometric (body height and weight) and routine office BP measures taken as per current guidelines[45]; PP was calculated as systolic BP − diastolic BP. Relevant comorbidities and ongoing medications were recorded. On the same occasion we administered a short questionnaire to estimate sodium intake, we measured TEWL and we collected a skin biopsy, serum and EDTA-plasma (for p-Na, p-Urea and p-Creatinine and for NT-proBNP, respectively) and a random urine sample (for albuminuria).

**Questionnaire**. A short, validated questionnaire[46] was administered to patients while waiting for their scheduled visit (Supplementary Note 2). Calculation was made as reported[46]. Briefly, it included 42 food items with six possible consumption frequency responses: never; one to three times per week; four to six times per week; once a day; twice a day; and three plus times a day. Predefined absolute amounts of sodium per serving size per specific item, according to *MRC Food Composition Tables*[47] were multiplied by the consumption frequency factor that each individual reported, and summed up to a total weekly $Na^+$ intake for each subject, later divided by seven to estimate daily intake. For simplicity, the absolute amounts of Na per serving for each food category were divided by 50 mg $Na^+$ units and rounded to the nearest integer[46]. The frequency factor in the weekly calculation was taken as the value midway between the upper frequency value of one category and the lower of the next (i.e., 0, 2, 5, 7, 14, 21).

Missing questionnaire data were missing completely at random (Little's MCAR test, $p = 0.593$). For five subjects, with more than half of the responses missing, calculations of weekly scores were considered unreliable and excluded. Twenty-two additional subjects had $\leq 10\%$ missing responses (10/22 had only 1 missing item): median substitution was used for imputation and calculation of weekly scores. Results without imputation were overall identical.

**Transepidermal water loss**. Before blood collection and skin biopsy and following a $\geq 20$-min period of acclimatisation in a temperature-controlled environment (20–21 °C) and $\geq 5$ min of quiet sitting, TEWL[37] was assessed from the flexor portion of the forearm. We used a Tewameter® TM300 probe (Courage & Khazaka GmbH, Cologne, Germany), which estimates TEWL through two pairs of temperature and humidity sensors and Fick's diffusion law. Readings were automatically stopped and recorded by an MPA 580 system and dedicated software when a $\leq 0.1$ standard deviation was reached.

**Skin biopsy**. Approximately 40 min before the planned biopsy, lidocaine-based topical anaesthetic cream (LMX4) was applied on the outer upper arm, approximately halfway between the elbow and shoulder; at variance with other commercially available preparations, this cream was found to be free of any detectable $Na^+$ or $K^+$ content at flame photometry analysis in a pilot study in 18 healthy volunteers, used for methods development (SKILLS; University of Glasgow, MVLS College Ethics Committee, ref. 200160109). After cleaning the skin with cotton gauze pads and $Na^+/K^+$-free 70% alcohol wipes, skin punch biopsies were performed on the anaesthetised site with a disposable instrument (3–4 mm blade diameter; Kai Medical). The sample was immediately put into a pre-cooled Eppendorf tube, frozen in dry ice and subsequently stored at −80 °C until tissue analysis.

**Human healthy subjects**. A preliminary cross-sectional study on young healthy volunteers (SOWAS: SOdium and WAter Skin Balance—Insights into Local Regulation) was approved by the University of Glasgow, MVLS College Ethics Committee (ref. 200170153) and conducted between July and December 2018. The study was conducted in compliance with the Declaration of Helsinki. After consent was obtained, normotension confirmed and comorbidities ruled-out during a preliminary screening visit, young (19–35 years old) participants recruited among MVLS students attended a morning visit (starting at 8.00–9.30) after fasting from midnight. For female participants, the date of the main visit was arranged in the early follicular phase of their menstrual cycle, just after period termination; in four cases, due to personal logistics or continuous progesterone treatment (skin implants or vaginal coil) the main visit coincided with a luteal or luteal-like phase. During the visit, we measured BP as previously described, we recorded body weight and bioimpedance-calculated body composition (Tanita, BC-418MA) and collected serum and EDTA-plasma samples. Sodium excretion on 24 h urine collection, initiated after the first micturition of the day before and terminated with the first on the morning of the visit, was assessed semi-quantitatively on site by colorimetric strips (Salinity View, Health Mate) and later by urinary $Na^+$ quantitation by ion-selective electrodes (ISE indirect Na–K–Cl Cobas, Roche Diagnostics GmbH, Mannheim) to exclude excessively high $Na^+$ consumption (i.e., >5 g/die), confirmed for all participants. With an anaesthetic and surgical protocol identical to what described above, with the exception of the biopsy site, we collected two adjacent 4 mm punch skin biopsies from the upper external gluteal quadrant. One of the samples was cut along a sagittal median plane on the bench to symmetrically include both epidermis and dermis in both halves, immediately put into a pre-cooled Eppendorf tube, frozen in dry ice and subsequently stored at −80 °C until tissue analysis.

**Human skin samples analysis**. Frozen skin samples were macroscopically transversally cut into a superficial layer, including the epidermis and the immediately adjacent superficial dermis (ESD), and DD layer in a cold room (please see Supplementary Fig. 7). Tissue processing was identical to what described for rat tissues, with the exception of a five decimal (0.00001 g) scale (Ohaus, DV214CD) used for weight measurements due to small size of the samples. Water and $Na^+/K^+$ contents were measured as described for rat tissues. The few missing values secondary to technical issues or insufficient sample were excluded.

Paraffin-embedded samples from the pilot studies mentioned above were used for histological pictures, representative of sex and age differences. After deparaffinization (Histoclear) and progressive rehydration, 5 µm sections including both epidermis and dermis were stained with Picrosirius Red (1 h) or, after overnight antigen retrieval at 40 °C in Unitrieve (Innovex Biosciences), with standard DAPI staining protocols.

**Statistics**. All data were collected in Microsoft Excel spreadsheets. Statistical analysis was performed using Prism (GraphPad Software) and SPSS (IBM).

Categorical variables are presented as absolute numbers and percentages and compared by $\chi^2$ test. The effect of two factors (e.g., strain and salt load) on different quantitative response variables in the experimental groups was tested by two-way ANOVA. For predefined comparisons (e.g., the effect of HS vs. NS) we used Fisher least-significant-difference test (animals) or Student's $t$ test (humans) for normally distributed variables (presented as mean ± SD or, graphically, as mean [95% CI]) or Mann–Whitney test for non-normally distributed variables (presented as median [interquartile range] or, graphically, as median [95% CI]). For post hoc comparisons (e.g., between strains or sexes), labelled as such in the relative figures/tables, Holm–Sidak test was used. Data tested against a specified value were analysed by one-sample $t$-test. Prior to any comparison, outliers were identified by ROUT method ($Q = 1\%$) and excluded from analysis but reported on the relative

figures. Regression curves were derived by least-square method and compared by Extra sum-of-squares $F$ test; for vascular function analysis, maximal contraction/relaxation and (Log)EC50/IC50 were independently compared. Correlations were ascertained by Pearson test, upon appropriate transformation of skewed variables to attain normal distribution. Univariable and multivariable (including age, sex, BMI and estimated Na intake) linear regression models were used and results were presented as standardised $B$ coefficients (95% CI). The $\alpha$ level was set at 0.05 and all statistical tests were two tailed (*$p < 0.05$, **$p < 0.01$, ***$p < 0.001$, ****$p < 0.0001$).

**Reporting summary**. Further information on research design is available in the Nature Research Reporting Summary linked to this article.

## Data availability

The authors declare that all data supporting the findings of this study are available within the paper and its supplementary information. Source data are provided with this paper.

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

## Acknowledgements

We gratefully acknowledge the support from Dr. Katriona Brooksbank (University of Glasgow, UoG) for protocol optimisation and ethical/NHS R&D approvals; from Adam Harvey for the rat studies; from Michael Beglan, Holly Yu and Prof. Samuel Jackson's team (UoG; Chemistry) for the chemical analysis of samples; from Jackie Thomson and Elaine Butler (UoG) for their technical assistance and processing of plasma and urine samples; from Kayley Scott (UoG), Laura Haddow and John McAbney (MyoCORE facility, UoG) for the animal protocols and myography experiments; from all the consultants in the Glasgow High Blood Pressure Clinic (Queen Elizabeth University Hospital) for the recruitment of participants. We are also grateful to Dr. Helen Taylor, EnviroDerm Services (UK distributor of Courage & Khazaka products), for facilitating our access to TEWL equipment and for her technical and scientific support. None of the individuals/companies acknowledged above received compensation for their contributions to this study. This work was supported by the British Heart Foundation (BHF)

Centre of Research Excellence Awards, RE/13/5/30177 and RE/18/6/34217 to RMT, C.D. and G.R.; the Academy of Medical Sciences-Newton International fellowship to S.M.; a Carnegie Trust Undergraduate Vacation Scholarship, VAC008890 to J.Y.C. and University of Glasgow Head of College Scholars' List Scheme Summer Studentship Award 2017/18 to K.S.C. R.M.T. is funded through a BHF Chair award (CH/4/29762). A.C.M. is supported by a Walton Fellowship (University of Glasgow).

## Author contributions

G.R. had full access to all of the data in the study and takes responsibility for the integrity of the data and the accuracy of the data analysis. Concept and design: G.R. and C.D.; experiment conduction and interpretation: G.R., S.M., J.Y.C., P.B., A.C.M., R.M.T. and C.D.; human study design, recruitment of participants samples/data collection: G.R., J.Y.C., K.S.C., R.M.T. and C.D.; performance, analysis and interpretation of myography data: G.R., K.B.N., R.L.A. and R.M.T.; analysis and interpretation of biochemical data: G.R., P.W. and C.D.; drafting of the paper: G.R. and J.Y.C.; critical revision of the paper for important intellectual content: G.R., S.M., J.Y.C., P.W., M.C.P., R.M.T. and C.D.; statistical analysis: G.R.; supervision: R.M.T., M.C.P. and C.D.; funding: G.R., S.M., J.Y.C., K.S.C., R.M.T. and C.D.

## Competing interests

The authors declare no competing interests.
