## [Peer Review File · Nature Communications]

Reviewers' comments:

Reviewer #1 (Remarks to the Author):

Gradients in Na concentration are observed in kidney: hypertonicity in the medulla vs isotonicity in the cortex. Elaborate arrangements of renal tubules in combination with specific enrichment of active sodium transporters (Na,K-ATPase, Na,K,Cl cotransporters, Na, Cl cotransporters, etc) are required for the sodium gradient in addition to enormous flow in tubules and blood vessels. On the other hand, structural basis for local hypertonicity in dermis or other non-renal tissues is not known casting doubts. This study provides a solid ground for the pessimism based on well performed comprehensive analyses of animals and patients. Data presented are all consistent with expansion of ECV which is systemic rather than skin specific. Given the lack of evidence for structural or functional features required for the maintenance of local hypertonicity in the skin, these data are conceptually more plausible. In the end, the study provides an important insight to the issue of non-renal local hypertonicity which has been plagued by conflicting reports.

By Hyug Moo Kwon

Reviewer #2 (Remarks to the Author):

The manuscript by Rossitto G. et al. showing that salt accumulation in the skin is not water-independent but isotonic nature is provocative and interesting. Moreover, excess of tissue Na⁺ upon high salt intake is not a skin-specific but systemic phenomenon reflecting an architectural change which is attributable to cellular loss and fat distribution. Results of these in vivo and ex vivo experiments in hypertensive rats and skin studies in hypertensive subjects are attractive and informative, but there are several defects to be resolved.

1. In SuppFig 8, at variance with aging, Na/K ratio was shifted to the right, despite the sentence "the lack of any significant shift in the Na/K ratio suggest a fluid accumulation"
2. According to the sentence "Office BP was inversely associated with ESD water ---- (Fig4e)", no data of diastolic BP in Fig4e.
3. Again, "No significant interaction with medications---, positively--, and systolic BP and negatively with diastolic BP, mirrored all measures of skin oedema accumulation (Fig4e)". But, no data of systolic and diastolic BP in Fig4e.

4. "Na⁺ and K⁺ were lower and higher in the ESD compared to DD, respectively (Fig4a)". However, the difference of Na⁺ between ESD and DD (mean+SD), as shown in the red columns, seemed very small. Data should be shown by dot. Water content in ESD and DD also be shown.

5. How do you explain about the opposite vascular responses to SNP between HS diet and hypertonic Na⁺ (Fig3b)?

6. In Fig4d showing water, Na⁺ and K⁺ in ESD with age, how about these parameters in DD?

7. Some investigators measured the interstitial fluid osmolarity, Na⁺, and K⁺ measured by microdialysis (Nikpey E et al. Hypertension 69: 660, 2017), and they showed the significant differences in either osmolarity or Na⁺ between interstitial fluid and plasma in both normotensive and hypertensive rats. Is there any difference in Na⁺ concentration between plasma and tissues?

Reviewer #3 (Remarks to the Author):

Add reference to TEWL literature (textbook and/or article), such as:

Joachim Fluhr, Peter Elsner, Enzo Berardesca and Howard I Maibach, Editors. Bioengineering of the Skin. Water and the Stratum Corneum, 2nd Edition. CRC Press, Boca Raton, FL, 2005

Reviewer #4 (Remarks to the Author):

I read carefully the part of manuscript on the measurements of potassium and sodium by flame photometry in tissues. this part is well prepared and the results are analytically acceptable .

Response to Reviewers' comments:

Reviewer #1:

Gradients in Na concentration are observed in kidney: hypertonicity in the medulla vs isotonicity in the cortex. Elaborate arrangements of renal tubules in combination with specific enrichment of active sodium transporters (Na,K-ATPase, Na,K,Cl cotransporters, Na, Cl cotransporters, etc) are required for the sodium gradient in addition to enormous flow in tubules and blood vessels. On the other hand, structural basis for local hypertonicity in dermis or other non-renal tissues is not known casting doubts. This study provides a solid ground for the pessimism based on well performed comprehensive analyses of animals and patients. Data presented are all consistent with expansion of ECV which is systemic rather than skin specific. Given the lack of evidence for structural or functional features required for the maintenance of local hypertonicity in the skin, these data are conceptually more plausible. In the end, the study provides an important insight to the issue of non-renal local hypertonicity which has been plagued by conflicting reports.

We would like to thank Rev#1 for the words of appreciation.

Reviewer #2:

The manuscript by Rossitto G. et al. showing that salt accumulation in the skin is not water-independent but isotonic nature is provocative and interesting. Moreover, excess of tissue Na+ upon high salt intake is not a skin-specific but systemic phenomenon reflecting an architectural change which is attributable to cellular loss and fat distribution. Results of these in vivo and ex vivo experiments in hypertensive rats and skin studies in hypertensive subjects are attractive and informative, but there are several defects to be resolved.

1. In SuppFig 8, at variance with aging, Na/K ratio was shifted to the right, despite the sentence "the lack of any significant shift in the Na/K ratio suggest a fluid accumulation"

We thank the reviewer for giving us the opportunity to reword the sentence and clarify: the "lack of any significant shift in the Na/K ratio..." referred to associations with estimated salt intake (third pair of columns in Supplementary Figure 8). With ageing, Na/K ratio was indeed shifted (as shown also in the representative histological pictures).

2. According to the sentence "Office BP was inversely associated with ESD water ---- (Fig4e)", no data of diastolic BP in Fig4e.

3. Again, "No significant interaction with medications---, positively--, and systolic BP and negatively with diastolic BP, mirrored all measures of skin oedema accumulation (Fig4e)". But, no data of systolic and diastolic BP in Fig4e.

We thank the reviewer for the comments. Data for systolic BP and diastolic BP were not included in figure 4 to avoid visual overcomplication, but are now detailed in the main text. Pulse pressure, which incorporates information from both systolic and diastolic BP and is an important measure of arterial stiffness in the hypertensive population (particularly in the elderly), features in Figure 4.

4. "Na+ and K+ were lower and higher in the ESD compared to DD, respectively (Fig4a)". However, the difference of Na+ between ESD and DD (mean+SD), as shown in the red columns, seemed very small. Data should be shown by dot. Water content in ESD and DD also be shown.

Based on Reviewer's suggestions, we have now included data for water, which are in keeping with the higher fat content in DD (and lower % weight of water, accordingly) compared to ESD, and we have changed the scale of the Y axis to allow better appreciation of the differences (bars presented as mean \pm 95%CI). We are happy to provide the dot plots here below:

however, in order to facilitate readability of the whole figure and avoid visual overcomplication, we would ideally prefer bars. Single dots are presented in panel B, to show that virtually no patient had $[Na^+ + K^+]$ exceeding physiological values.

5. How do you explain about the opposite vascular responses to SNP between HS diet and hypertonic Na+ (Fig3b)?

We thank the Reviewer for the opportunity to further strengthen or conclusion: because the two experimental conditions are not reflecting the same phenomenon or, in other words, because HS diet does not result in a perivascular hypertonic environment. As we suggest in the manuscript, the “paradoxical” responses observed with hypertonic Na⁺ are not reminiscent of the vascular changes usually sustaining hypertension – where relaxation, if at all, is impaired. The effects observed after incubation with hypertonic Na⁺ are more reminiscent of conditions where neuromuscular tissues are exposed to a supraphysiologic Na⁺ concentration, “toxic” to the cell membrane electrochemical equilibrium and resulting in loss-of-function (e.g. lethargy or muscle weakness), like diabetes insipidus.

6. In Fig4d showing water, Na+ and K+ in ESD with age, how about these parameters in DD?

DD chemical analysis is affected by dermal fat content, which impacts on water, Na⁺ and K⁺ content by limiting their volume of distribution. Therefore, differences in dermal fat content (as observed between males and females; Supplementary Table 4 and Supplementary Figure 7) as well as unavoidable microscopic differences in the depth of the surgical biopsy procedure are more difficult to account for in DD analyses and give justification as to why ESD is more ‘reliable’. Nevertheless, a shift in Na/K ratio with ageing (Supplemental Figure 8) is evident also in DD despite these technical limitations.

7. Some investigators measured the interstitial fluid osmolarity, Na+, and K+ measured by microdialysis (Nikpey E et al. Hypertension 69: 660, 2017), and they showed the significant differences in either osmolarity or Na+ between interstitial fluid and plasma in both normotensive and hypertensive rats. Is there any difference in Na+ concentration between plasma and tissues?

Nikpey et al tried multiple methodological approaches to measure interstitial fluid osmolarity and Na⁺. In fact, despite an increase in skin Na⁺ (and water, very consistently with our findings reported

in Figure 2 and Supplementary Figure 2), they concluded that “Interstitial fluid isolated from implanted wicks and lymph draining the skin was isosmotic, and Na⁺ concentration in fluid isolated by centrifugation and in lymph was not different from plasma” (i.e. isotonic). As the increased skin Na⁺ could more easily be explained by a shift in ECV/ICV ratio than by a supposed phenomenon that couldn’t be captured, we interpret their findings as further supporting the lack of interstitial hypertonicity upon high Na⁺ diet.

Unless a (theoretical) tissue is composed exclusively of extracellular volume (i.e. ECV% = 100% in Supplementary Figure 2a), tissue Na⁺ concentration will always be lower than plasma Na⁺ concentration (≈140 mM): in the whole tissue homogenate required for tissue Na⁺ measurement, Na⁺ concentration will be “lowered” by the fraction of the tissue which is poor in Na⁺, i.e. the intracellular (please see Figure 1). The more ECV% approximates 100%, the more tissue Na⁺ approximates ≈140 mM (and vice versa): this is clearly shown in Figure 4 a, where tissue Na⁺ in ESD (richer in cells) is lower compared to the mostly extracellular DD.

Reviewer #3

Add reference to TEWL literature (textbook and/or article), such as:

Joachim Fluhr, Peter Elsner, Enzo Berardesca and Howard I Maibach, Editors. Bioengineering of the Skin. Water and the Stratum Corneum, 2nd Edition. CRC Press, Boca Raton, FL, 2005

We thank the reviewer for correctly pointing this omission. The suggested reference has been included in both main text and supplemental material.

Reviewer #4:

I read carefully the part of manuscript on the measurements of potassium and sodium by flame photometry in tissues. this part is well prepared and the results are analytically acceptable.

We would like to thank Rev#4 for the appreciation.

REVIEWERS' COMMENTS:

Reviewer #2 (Remarks to the Author):

With the appropriate responses to the reviewer's comments, the authors have carefully modified their manuscript.